# Global Wheat Spike Detection and Quantitative Analysis Based on Image Recognition (Proposal)

**Xun Wang**
Institute for Interdisciplinary Information Sciences
2024311552
wang-x24@mails.tsinghua.edu.cn

**Tianhai Liang**
Institute for Interdisciplinary Information Sciences
2024211278
liangth24@mails.tsinghua.edu.cn

**Gu Zhang**
Institute for Interdisciplinary Information Sciences
2024311577
zg24@mails.tsinghua.edu.cn

## Abstract

Wheat is one of the most important staple crops worldwide. Rapid and accurate detection of wheat spikes from outdoor field images is essential for farmers to implement efficient field management and enhance the quality of wheat cultivation. However, factors like planting environment and image quality complicate achieving robust and precise detection using visual models. Therefore, we propose a framework that employs adversarial training with diffusion models and YOLO-v11 visual models to address this challenge.

## 1 Background

Wheat ranks among the most critical staple crops worldwide, extensively cultivated and providing a primary food source for a significant global population.[Senapati et al., 2022] Thus, increasing the efficiency and quality of wheat production is vital for raising living standards and ensuring social stability. To support farmers in monitoring wheat growth and making informed field management decisions, an efficient approach is to quickly detect and analyze the number and size of wheat spikes using field images. However, developing a visual system capable of accurately identifying and quantitatively analyzing wheat spikes under varying global conditions is challenging, given the diversity in wheat varieties, environmental differences, and potential issues like spike overlap and image blurring. Such a system, if successful, could greatly aid farmers in optimizing wheat production while advancing research in agricultural and food sciences.

## 2 Definition

We formalize the global wheat spike detection and quantitative analysis problem as an object detection task. Let $\mathcal{I}$ denote the set of all possible images. A sample of the problem is defined as a pair $(I, Y)$, where $I \in \mathcal{I}$ represents a field image containing various quantities of wheat spikes, and $Y = \{(x_i, y_i, w_i, h_i)\} \subset \mathbb{R}^4$ is the set of bounding boxes for the wheat spikes in image $I$.

Preprint. Under review.

For this task, we utilize a training dataset $\mathcal{D}_{train} = \{(I_i, Y_i)\}_{i=1}^{N_{train}}$, where $I_i$ is the $i$-th wheat image and $Y_i$ contains the associated true bounding boxes. The objective is to learn a parameterized model capable of accurately detecting the positions of wheat spikes in a test dataset $\mathcal{D}_{test} = \{(I_j, Y_j)\}_{j=1}^{N_{test}}$, where the sample structure mirrors that of the training dataset. Specifically, the model is implemented as a neural network $f_\theta : \mathcal{I} \to \mathcal{P}(\mathbb{R}^4)$, with $\theta$ representing the parameters and $\mathcal{P}(\mathbb{R}^4)$ being the power set of $\mathbb{R}^4$. The output for the $j$-th image is the detected bounding boxes $\hat{Y}_j = f_\theta(I_j) = \{(x_k, y_k, w_k, h_k)\}_{k=1}^{N_j}$, where $N_j = |\hat{Y}_j|$ is the estimated quantity of wheat spikes in the input image.

To evaluate the detection model's performance, we employ the Intersection over Union (IoU) metric as the threshold and calculate the average precision over a range of IoU thresholds, following the Kaggle competition guidelines [David et al., 2020b].

# 3 Related Work

Several vision methods have been employed for the wheat spike detection task. For instance, Zang et al. [2022], Zhao et al. [2022] and Guan et al. [2024] utilized YOLO models, a powerful series of frameworks known for their expertise in object detection [Hussain, 2024, Wang et al., 2024]. In contrast, Wen et al. [2022] developed their vision models based on RetinaNet, another object detection framework that features Focal Loss [Lin, 2017]. Additionally, Zhou et al. [2022] detected wheat spikes using models adapted from the Swin Transformer architecture [Liu et al., 2021]. Beyond these approaches utilizing off-the-shelf backbones, Ye et al. [2024] concentrated on creating a novel network framework tailored specifically for this task. While these methods have demonstrated good detection accuracy, they give little consideration to generative models for adversarial training, despite the potential of such architectures to further enhance the model's accuracy and robustness.

# 4 Proposed Method

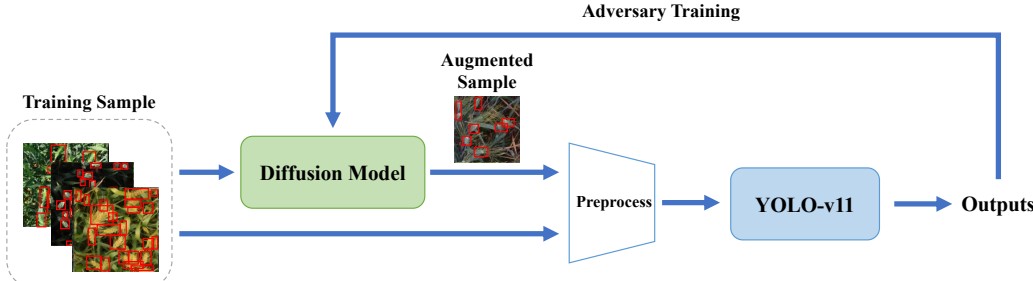

Figure 1: The framework of our proposed method.

Building on existing research and considering computational costs, we plan to address the problem using specific pretrained vision models and propose a more effective fine-tuning pipeline. First, to enhance the generalization ability of our model, we will employ generative models for data augmentation. For instance, the diffusion model [Croitoru et al., 2023] is a popular and effective choice for fitting the distribution of training samples and generating diverse augmentations. Next, we will perform preprocessing on the raw images from both the training set and the diffusion model, similar to other computer vision tasks. Finally, we will utilize YOLO-v11 [Jocher and Qiu, 2024], a cutting-edge object detection model, to achieve precise wheat spike detection.

To better leverage generative capabilities and enhance performance, we will implement an adversarial framework to train our models. Our approach involves fine-tuning the models over several epochs, alternating between training the generative model and the vision model in each step. This strategy is expected to enable the generative model to produce more challenging samples, thereby improving the detection capabilities of the model. An illustration of our proposed method is shown in Figure 1.

To evaluate our method, we plan to compare it with baseline models, such as pure YOLO, using the Global Wheat Head Detection (GWHD) dataset [David et al., 2020a].

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
