# OpenReview forum: "[Proposal-ML] Global Wheat Spike Detection and Quantitative Analysis Based on Image Recognition"
_tsinghua.edu.cn/THU/2024/Fall/AML — THU 2024 Fall AML Submission_

### Official Review · ~Shaoting_Zhu1 · 2024-11-06
**Review of submission 39**

**Rating:** 10
**Confidence:** 4

**Review:**

The proposal presents a framework aimed at detecting and quantifying wheat spikes from outdoor field images using advanced image recognition techniques. This is a critical task for agricultural management, as it can aid in optimizing wheat production and enhancing the quality of cultivation. The authors propose a sophisticated approach that combines adversarial training with diffusion models and YOLO-v11 visual models to achieve robust and precise detection under varying conditions.

**Stength**
1. Innovative Framework: The integration of adversarial training with diffusion models and YOLO-v11 is a novel approach that could potentially enhance the robustness and accuracy of wheat spike detection.
2. Clear task and metrics: The authors plan to evaluate the model using the Intersection over Union (IoU) metric and following Kaggle competition guidelines. The **Definition** paragraph clearly defines the task. The dataset and pre-trained model are clearly surveyed in the proposal, making the project very feasible.

**Weakness**
To train a diffusion model and a yolov11 model together may cost a lot of computation resources.

Overall, this proposal is well-written, and the task, dataset, and method are very clear.

---

### Official Review · ~Lily_Sheng1 · 2024-11-08
**Submission 39 Review**

**Rating:** 10
**Confidence:** 4

**Review:**

This work proposes an advanced wheat spike detection approach by leveraging a combination of generative models for data augmentation and a YOLO-based object detection. The method incorporates adversarial training to generate challenging augmented samples to improve detection accuracy and generalization.

Pros:
1. There are clear approaches and baselines defined.
2. Using a diffusion model for data augmentation helps to improve the model's robustness and generalizability.
3. This work aligns well with the practical needs of agricultural monitoring.

Cons:
1. This approach may heavily depend on the quality of generated images, which can vary depending on the diffusion model's capability to accurately replicate real-world visual diversity.

---

### Official Review · ~Bryan_Constantine_Sadihin1 · 2024-11-09
**Review of "Global Wheat Spike Detection and Quantitative  Analysis Based on Image Recognition"**

**Rating:** 10
**Confidence:** 5

**Review:**

Strength:
1. High Relevance and Potential Impact: The proposal offers a solution for improving wheat production, which is important for global food security.
2. Use of advanced method: The proposal clearly defines the underrepresented class problem in wheat detection, and proposes uses generated augmented samples.

---

### Official Review · ~Matteo_Jiahao_Chen1 · 2024-11-10
**Well-structured proposal fo Global Wheat Spike Detection and Quantitative Analysis**

**Rating:** 9
**Confidence:** 4

**Review:**

This work proposes a framework that combines diffusion-based data augmentation with YOLO-v11 in an adversarial training setup to improve wheat spike detection in field images.
### Strengths
-  The combination of diffusion models with YOLO-v11 in adversarial training may improve detection robustness but it may depend on the quality of the generated images.
- The framework addresses real-world challenges  benefiting agricultural management.

### Weaknesses
- A deeper analysis of model performance under varied field conditions would strengthen the study.

---

### Official Review · ~Qihang_Cen1 · 2024-11-12
**Practical research problem, clear technical framework**

**Rating:** 9
**Confidence:** 4

**Review:**

This paper focuses on global wheat spike detection and quantitative analysis, which is a well pratical problem and has significant implications for global food security. Furthermore, the study offers a technique framework using GANs combined with YOLO and diffusion models for data augmentation, with adversarial training to increase the robustness. This framework is innocative and well-structured, with clear details about model and experiment present in proposal. Additionally, maybe append discussion the advantages of model selection could provide readers with more clarity on its suitability.

---

### Official Review · ~Kaiyuan_Zhang6 · 2024-11-12
**Good proposal**

**Rating:** 9
**Confidence:** 5

**Review:**

A clear proposed topic and relative fulfilled description on background, definition, related work and methods, and include technique details. Besides, the methods is clearly presented via a framework figure.
One possible challenge may be how to guarantee the augmentation data using diffusion model is still a wheat picture, which should be discussed in the future work.

---

### Official Review · ~Zhixuan_Pan1 · 2024-11-12

**Rating:** 9
**Confidence:** 4

**Review:**

This project aims to improve global wheat spike detection using image recognition. By combining adversarial training with diffusion models and the YOLO-v11 architecture, the project seeks to achieve accurate and robust detection across diverse environmental conditions.

Pros:

1. Combining diffusion models and YOLO-v11 for adversarial training framework like GAN is a novel method.

2. The proposal's objectives are specific. The methods presented are concrete and detailed.

Cons:

1. More ablation studies may be needed, such as conducting experiments using only the diffusion model for data augmentation without adversarial training.

2. There is limited novelty in the methodology. It is similar to  many previous works in image classification.

---

### Official Review · ~Yuji_Wang4 · 2024-11-12
**Review of "Global Wheat Spike Detection and Quantitative Analysis Based on Image Recognition"**

**Rating:** 9
**Confidence:** 3

**Review:**

The project focuses on the problem of global wheat spike detection. The authors model the problem as image recognition and propose to solve global wheat spike detection problems with a combination of diffusion models and object detection models, which will be trained with an adversarial method.

### Strengths:

1. Method design: The proposal offers a framework that leverages generative models (diffusion models) to improve object recognition accuracy.
2. Feasibility：The research problem is well-defined, and related works are thoroughly discussed. The experimental plan is concrete, making the project feasible.

### Weaknesses:

1. Clarity of writing: The proposal lacks specific details on how to integrate diffusion models and adversarial training with object recognition models.

Concerns: Is there evidence (e.g., difficulty of the problem, limitations of existing methods) to justify the need of applying the sophisticated pipeline to the task? Will the use of adversarial training introduce training instability?

---

### Official Review · ~Bowen_Gao1 · 2024-11-12
**Review of Global Wheat Spike Detection and Quantitative Analysis Based on Image Recognition**

**Rating:** 9
**Confidence:** 4

**Review:**

**Summary**

This proposal addresses the Global Wheat Spike Detection task, where the authors propose using adversarial training to tackle challenges related to diverse planting environments and varying image quality.

**Strengths**

1. The problem definition and proposed method are clearly stated, supported by mathematical formulations that enhance clarity and rigor.
2. The review of related work is comprehensive
3. The inclusion of a framework figure effectively illustrates the entire pipeline, making it easier to understand the methodology and workflow.

**Weaknesses**

1. The explanation of challenges in previous methods could be expanded, providing more insight into the specific limitations and issues that the proposed approach aims to overcome.

---

### Official Review · ~Chendong_Xiang1 · 2024-11-12
**interest topic**

**Rating:** 8
**Confidence:** 2

**Review:**

This paper proposes a framework combining diffusion models and YOLO-v11 with adversarial training for global wheat spike detection and quantitative analysis. The approach enhances model generalization and robustness across diverse environments through data augmentation and adversarial sample generation. The authors plan to validate the framework on the Global Wheat Head Detection (GWHD) dataset, aiming to support effective wheat growth monitoring in agricultural production.